# Investigating the long-term impact of misinformation interventions in upper secondary education

**Thomas Nygren** *, **Evgenia Efimova**

Department of Education, Uppsala University, Uppsala, Sweden

* thomas.nygren@edu.uu.se

## Abstract

This study examines the long-term effects of educational interventions aimed at improving upper-secondary students' ability to identify and evaluate (mis)information. Using an experimental design with 459 students in authentic classroom settings, we tested three types of interventions: prebunking through the *Bad News* game, fact-checking skills via the *News Evaluator* workshop, and subject-specific disciplinary literacy interventions. Additionally, we investigated factors influencing students' abilities to identify credible and misleading news, including credibility importance, democratic ideals, actively open-minded thinking (AOT), self-rated fact-checking skills, and educational orientation.

Neither the *Bad News* game nor the *News Evaluator* workshop significantly improved students' ability to evaluate credible and misleading news. Although students in the *News Evaluator* workshop slightly improved in evaluating a difficult-to-fact-check item, this effect was marginal. Furthermore, students did not report increased use of digital verification tools three months after the intervention, suggesting that isolated fact-checking instruction may be insufficient for fostering lasting behavioural change. The subject-specific interventions also failed to yield significant long-term effects.

Students who rated access to credible information as important were better at identifying accurate news, while those who valued democracy were better at recognising false news. AOT was not associated with better discernment but increased scepticism toward both true and false news. Self-rated confidence in news evaluation did not predict actual performance, and students in vocational programs performed worse in identifying trustworthy news.

While interventions promoting prebunking, fact-checking, and disciplinary literacy provide promising frameworks for misinformation education, their effects may diminish without sustained reinforcement. This study underscores the need for more embedded, repeated, and adaptive approaches that integrate retrieval practice, spaced learning, and digital tools to foster long-term engagement with credible information.

**Data availability statement:** Data and R code are available at: https://osf.io/fnp8u/files/osfstorage.

**Funding:** This study was funded by the Swedish Institute for Educational Research, grant no 2020-00009. Funding awarded TN The funders had no role in study design, data collection and analysis, decision to publish, or preparation of the manuscript.

**Competing interests:** The authors have declared that no competing interests exist.

Future research should explore how to enhance the impact of classroom-based misinformation education and investigate scalable strategies for reinforcing digital source evaluation skills over time.

## Introduction

Researchers and international organisations call for more and better education, focusing on safeguarding democracy against the challenges of misinformation, i.e., the intentional and unintentional spread of misleading information [1,2]. It has been noted that educating young people to navigate complex media environments better is possible. Still, only a limited number of educational interventions have been tested and evaluated scientifically in ordinary classrooms [1,3]. Inoculating young people by exposing them to short game-based interventions, so-called prebunking [4], as well as interventions teaching students how to fact-check digital news more like an expert, so-called lateral reading [5,6], may have a significant impact in classroom settings to strengthen students abilities to evaluate (mis)information. This study investigates long-term effects of misinformation interventions, an area that remains under-researched in classroom settings.

Even if research today presents some promising tools for individuals to use against misinformation [7], there are still many challenges. Most students struggle to separate credible news from misinformation [8–10]. Students may also be overconfident about their ability to navigate misinformation [11]. Interventions with significant impact in classrooms hold important limitations. All educational interventions so far have an impact of Cohens $d = 0.7$ or lower, and the impact is limited even when researchers are involved in the education design. For instance, Wineburg et al. [6] note that short classroom interventions cannot "do magic." In addition, Maertens et al. [12] identify a lingering effect from lab interventions if participants do not get additional boosts, and there is a lack of research investigating the long-term effect of classroom interventions. Further complicating the challenge is the risk of interventions against misinformation unintentionally making people excessively critical of legitimate news [13–15].

### The current study

Previous research has provided various definitions of literacies related to the ability to navigate (mis)information [16]. In this study, we use the term media and information literacy (MIL) as an umbrella term that refers to people's knowledge, skills, and attitudes of importance to use online information in critical and constructive ways [17,18]. It includes the knowledge to find good information, skills to create new information in different formats, and critical evaluation of information found in different formats, such as texts and images [10, 19–22]. We focus specifically on students' abilities to identify and evaluate credible and misleading digital news and to what extent different educational interventions may have long-term effects on such abilities. While the terms "true," "credible," and "reliable" are often used interchangeably, we acknowledge their conceptual differences. In this study, reliable refers to a

receiver's perception of trustworthiness, which may not always align with the factual accuracy of credible news. In line with a Scandinavian tradition, we label this skill digital source criticism, which today needs to include the ability to evaluate information in updated ways in a digital world of information disorder [23,24]. It also relates to what Lewandowsky et al. (25) called "technocognition," a concept that underlines the importance of citizens understanding journalistic principles and how to use cognition and technology in updated ways to navigate the complicated landscape of (mis)information [25–27].

The primary purpose of this study is to investigate different approaches to stimulate digital source criticism within a classroom setting. It employs an experimental design in an authentic upper secondary school setting to test the long-term effects of three types of interventions against misinformation highlighted as important in previous research: inoculation (prebunking), fact-checking skills (lateral reading), and disciplinary literacy (subject-specific teaching). The inoculation and fact-checking interventions used well-researched instruments, the Bad News game [4,28,29] and the News Evaluator workshop [5,30]. In contrast, the subject-specific workshops were designed in cooperation with practising teachers during the previous stages of the project, with promising results indicating some short-term effects [31]. Additionally, this study looks into attitudes and other factors that might help explain differences in students' abilities to identify true and false news online, as such relationships might be important to bear in mind when designing and evaluating misinformation interventions.

## Theoretical background

Navigating and evaluating digital information is not intuitive and does not come naturally. To stimulate critical and constructive thinking in digital environments, theories linked to empirical studies highlight three perspectives of particular value to this project. The supporting theories emphasise the importance of (a) encountering disinformation in a diluted form to be inoculated against it, (b) learning lateral reading from fact-checkers, and (c) good subject-specific knowledge.

### Prebunking for inoculation

Prebunking is a psychological intervention strategy designed to build resistance to misinformation before individuals encounter it [28]. It works by exposing people to weakened or simplified versions of misleading arguments, manipulative tactics, or false claims in a controlled environment, allowing them to develop cognitive immunity against real-world misinformation. This approach is rooted in inoculation theory [32], which suggests that just as vaccines introduce a weakened form of a virus to stimulate immune defences, exposing individuals to misinformation techniques in advance helps them recognise and resist misleading content when they encounter it in the future. Through games developed for this purpose, the researchers see opportunities to help students and citizens better detect false and biased information in digital environments [28,33,34].

Advocates for prebunking interventions against misinformation emphasise the value of learning to identify and expose propaganda and disinformation to, for instance, help people critically and constructively engage with climate denialism and anti-democratic online activities [35,36]. Studies with online players internationally and in Sweden show that the *Bad news* game can contribute to a more critical approach to disinformation in the form of manipulative messages [28]. In classroom studies based on these theories, games in education have also shown effects on students' ability to identify manipulative strategies [4,37].

### Fact-checking, lateral reading and technocognition

Other researchers highlight the importance of teaching students to evaluate online information like professional fact-checkers. Fact-checkers often work laterally in digital environments and read laterally on different web pages to determine credibility in digital news feeds [38]. Using lateral reading to fact-check information involves critically examining (a) the sender, (b) the evidence, and (c) comparing the information with independent sources like fact-checkers and journalists do [6]. This research is based on theories that people use different heuristics–cognitive rules of thumb–to navigate

complex digital worlds [39]. Fact-checkers have been shown to have specific heuristics that allow them to quickly and accurately determine the credibility of digital information [38].

For Swedish upper secondary students, it may be challenging to review digital information this way [10], especially those who think they are good at digital source criticism [11]. Interventions in ordinary classrooms to get students to think and act more like fact-checkers have been shown to support their ability to evaluate digital information [6,30,38,40,41]. Such interventions also note how students need to better understand how to use digital tools, more like fact-checkers, to evaluate digital news and misinformation better. Thus, using digital tools to fact-check information can be an indication of technocognition and something to strive for in education [25–27].

## Disciplinary literacy

Researchers who emphasise disciplinary perspectives and what is called "disciplinary literacy" argue that source criticism is about thinking patterns and habits that need to be learned and practised in combination with subject knowledge [42]. This perspective, which does not contradict the abovementioned ones, emphasises that subject knowledge is closely linked to critical thinking [43], making it central to link "vaccination" and lateral reading to teaching different subjects [10].

Previous research in the field of "disciplinary literacy" has highlighted how education and expertise can play a role in dealing with manipulative digital information [44,45]. A particular challenge that has been recognised is that critical thinking is closely linked to education and subject knowledge [43,46]. It is easier to search, select, and respond to information if you have knowledge of the area you are exploring [47]. Lateral reading may lead you down a rabbit hole if you do not have proper knowledge and know where to find trustworthy information [21,48]. Swedish subject didactic research indicates that students' critical thinking in history, mathematics, physics, and Swedish is associated with their subject-specific knowledge, and students' skills in reviewing and evaluating information may differ between subjects [43]. The most important thing for good scores on national test questions assessing critical thinking in different subjects appears to be good subject knowledge [43]. Lurie & Mustafaraj [49] note that due to a lack of domain-specific knowledge, students may get lost online when they engage in lateral reading, and more current research highlights how lateral reading without proper knowledge about current events can misinform people since they end up with "evidence" from low-quality sources [48].

Students from different secondary school programs and adults with different educational backgrounds have shown significant differences in their ability to assess the credibility of news. For example, students in the aesthetics program have been particularly good at critically examining racially coloured news, and students in the science program have shown a better ability to evaluate disinformation about global warming [10,11]. Different types of challenges regarding the evaluation of digital information have become increasingly apparent in recent research [8,9,50]. However, what the differences are due to and how teaching can help students become better at digital source criticism in different subjects is unclear.

## Research questions and hypotheses

Noting the potentials and challenges of education to promote student source critical thinking, we investigate the following research questions:

RQ1: What are the long-term effects of three educational interventions—prebunking, fact-checking, and disciplinary literacy—on students' abilities to evaluate true and false news?

RQ2: Which factors (e.g., credibility importance, democratic ideals, AOT, confidence, educational track) are associated with students' abilities to evaluate true and false news?

In line with our research questions and our preregistration (S1 File), we test the following hypotheses. All hypotheses regarding impact focus specifically on delayed effects, assessed through a follow-up conducted three months after the interventions:

(H1) Students playing the Bad News game (inoculation intervention) will become significantly better at identifying misleading and credible headlines [4].

(H2) Students who participate in the News Evaluator workshop (fact-checking skills intervention) will become significantly better at identifying misleading and credible digital news [5,30].

(H3) Students who participate in the News Evaluator workshop will more frequently use digital tools to assess the credibility of news [5,30].

Given the lack of previous research on subject-specific interventions against misinformation, this study also explores their possible effects with an expectation that taking part in some of the five subject-specific workshops will make students better at identifying true and false news.

Finally, the study also investigates some of the potentially important factors related to students' abilities to evaluate (mis)information. We plan to replicate the positive effect of (H4) credibility importance [10,11], (H5) importance of democracy [51] and (H6) actively-openminded thinking [52], and also the negative effects of (H7) self-rated confidence when evaluating digital news [11] and (H8) studying in a vocational track [10].

## Interventions

**Bad news and News evaluator.** The intervention structure was tailored to integrate into a single lesson period, dedicating around half an hour to playing the Bad News game or using the News Evaluator workshop before transitioning into a discussion led by the teacher. Both interventions were conducted in line with previous research showing the impact of the Bad News game [4] and the News Evaluator [5,30] a few days after the interventions. This approach emphasised the essential contribution of active engagement from teachers in augmenting the educational value derived from using digital resources [26,40,53].

Following the intervention, teachers guided a conversation about the students' experiences, utilising question prompts outlined by the researchers in conversations with experienced teachers. After playing the game, they were asked what it was like to manipulate other people, what strategies they used for amassing followers, identify the six strategies, discuss which of the strategies they recognised from social media, who usually makes use of such strategies and if they believe any strategy to be more effective to deceive people. After the News Evaluator workshop, teachers asked the students to consider if any news items were particularly complicated to assess, what similarities and differences they could identify between the news in the test and their own news feeds, and how one can stay updated with credible news.

## Subject-specific interventions

The subject-specific interventions were designed in collaboration with teachers and researchers and tested in pretest and posttest interventions in different upper-secondary Swedish schools. The development of the interventions used two design cycles, with a first small-scale test of the materials in two groups of students, followed by evaluations and redesign for a second cycle with at least two hundred upper-secondary students with different backgrounds. All interventions included a guide for teachers, PowerPoint presentations, and classroom materials designed to stimulate the students' digital source critical thinking. The design was supposed to be subject-specific and complement more general interventions promoting inoculation and fact-checking abilities; links to all teaching materials can be found in S1 File.

In the social sciences, the intervention focused on analysing who is responsible for the truth, using Swedish free speech legislation and the Capitol storming incident as a problem. The teaching included comparing Swedish, US, and Hungarian free speech legislation and media types, discussing the balance between freedoms, protection of rights, and societal and national security, and addressing accountability for false information. Moreover, critical source assessment and lateral reading exercises were used to highlight the significance of diverse media actors and norms in safeguarding society from harmful information dissemination.

The intervention in history had a focus on making students reflect upon the heated online debate about Linnaeus being a racist and whether or not monuments of him should be removed. The educational design included a short lecture about Linnaeus showcasing different online posts about his legacy, lateral reading exercises and close reading of various contemporary and historical sources.

In psychology, the intervention focused on educating students to see different perspectives on critical thinking and what hinders us from being constructive critical thinkers. Over two and a half hours of instructions, students listened to short lectures, did charades, analysed sources, and watched videos. The aim was to help them see emotional, cognitive, and social psychological perspectives and how these all may influence our source critical thinking.

In science, the teaching focused on the scientific method, including peer review and scientific consensus. Content included visual illusions, data on COVID-19 death rates, contrasting views on vaccine side effects, information from the industry versus science about fossil fuels, and accurate and misleading information about climate change. The teaching also included exercises to support students' lateral reading skills to review and corroborate sources and highlighted the importance of understanding and trusting credible sources in the scientific community.

In art, the educational design was focused on educating students to (a) identify different types of images and understand their levels of manipulation, (b) critically analyse images, including reverse image search and simple image analysis, (c) consider the context and source of the image, such as its purpose, forum, and publisher, (d) compare the image with other sources and assess possible bias, cropping, or manipulation, and (e) create and manipulate images to understand the process and its implications.

## Power analysis

Previous studies showed that the Bad News game has an effect of $d = 0.3$ based on comparisons to the individual pretest (4) and $d = 0.7$ based on comparison to a control group [54]. News Evaluator has been shown to have an effect size of $d = 0.65$ [5]. Although these effect sizes come from studies with an immediate posttest, we expected that the school setting could provide the boost needed to support these effects long-term. To do some approximate power estimation for the cluster-randomised design, we used the PowerUp! tool [55]. This estimation suggested that the planned total sample of 1000 students (50 classes) could give a minimal detectable effect size of 0.33 (see Table A1 in Appendix A).

## Participants and procedures

The study included all classes in the school of a mid-sized Swedish public upper secondary school. Data was collected in December 2023 (pretest) and April 2024 (posttest), and students gave their consent digitally at the beginning of the questionnaires. As the participants were minors, informed consent was obtained from the students. The Swedish Ethical Review Authority approved the study and waived the requirement for parental or guardian consent (Dnr 2021−01340). All participation was voluntary, stating that students' grades would not be affected if they chose not to participate. All participants were informed of the correct responses to the test items after the data collection to ensure that the study would not support the unintended spread of misinformation.

For the inoculation and fact-checking interventions, existing classes were assigned to conditions by the school administration following the instructions provided by the first author. First, classes were randomly divided into two halves on a program-by-program basis so that each half would include a similar mix of study programs. Next, six classes from each of the halves were picked out randomly to make up a control group. The result was three non-overlapping groups: (a) Bad News game, (b) News Evaluator workshop, and (c) control. It was, however, not possible to properly control the randomisation process, which does not allow us to classify this study as fully randomised. For the subject-specific interventions, classes were assigned to participate in none to three of the five subject workshops based on students' scheduled teaching and voluntary participation.

The interventions to support inoculation and fact-checking were conducted in December 2023, and the subject-specific interventions were conducted between January and March 2024. The first author gave information about how to conduct the interventions to all teachers on-site at the school and provided them with short teaching manuals.

Out of the 50 classes that were supposed to participate in the study, seven did not find time in their schedules for the posttest or filled out the questionnaire without the correct codes. After additionally excluding duplicates and students without

consent or individual code, the pretest data has 718 students, and the posttest has data from 604 students. The resulting matched sample has 459 students from 43 classes. The median number of students in each class is 10, the minimum is 3, and the maximum is 23. In the matched sample, 198 students (15 classes) played the Bad News game, 182 students (19 classes) used the News Evaluator, and 79 students (9 classes) were in the control group. Table 1 provides an overview of the sample for the inoculation and fact-checking interventions, showing that students in the control group are, on average, more advanced in their studies ($\chi^2(4) = 16.13$, $p = 0.003$) and have a stronger academic background ($\chi^2(2) = 12.63$, $p = 0.02$). Although students in the control group were more advanced in their schooling and enrolled in more theoretical programs, this did not result in significantly higher pretest scores compared to the intervention groups. This suggests that more advanced academic placement does not necessarily translate into superior identification of misinformation or better fact-checking skills. Sample characteristics for the subject-specific interventions can be found in Table B1 in Appendix B.

## Measurements

Students' ability to evaluate true and false news was operationalised in relation to the different interventions within the study. For the Bad News game, we used four items similar to previous research on the effects of this intervention [4,28]. Each item showed a tweet with a headline, and students were asked, "How reliable do you believe this tweet to be?" They were asked to rate the reliability on a scale from one (very unreliable) to seven (very reliable). Three tweets were misleading and employed misinformation strategies that students could recognise from the intervention (imposter, discredit, and conspiracy), while one was credible and came from the Swedish public service. Given the low values of Cronbach's alpha for these four items, they were used separately in the analysis.

For the News Evaluator, we again employed two closely matched items in line with previous research on this intervention [30]. Both items asked the students to rate the reliability of Facebook posts on the same scale from one to seven. The posts included a picture and text that were searchable so that students could potentially implement their fact-checking skills from the intervention. The two items were used separately as outcome measures. Additionally, to capture lateral reading, students were asked if they used any digital tools: "Did you use any aids when answering the previous questions? Did you use, for example, Google or reverse image search?".

A broader approach was used for the subject-specific workshops and the analysis of factors related to the evaluation of true and false news. It combined tweets and Facebook posts, as well as four items with two credible and two misleading headlines inspired by the MIST instrument [56], which again were rated based on their reliability from one to seven.

Given the lack of internal consistency in any potential scale, the items were divided into true and false categories based on their meaningful real-life distinction between reliable and misleading news, even though Cronbach's alpha values were low (0.39 for true items and 0.54 for false).

Factors influencing students' ability to identify true and false news were measured using questions adapted from previous studies. The perceived importance of access to credible news, credibility importance [10,11], was assessed with the

**Table 1. Characteristics of the sample for inoculation and fact-checking skills interventions.**

| Variable | Control | Bad News | News Evaluator |
|---|---|---|---|
| Vocational track | 29 (37%) | 80 (42%) | 102 (56%) |
| Grade 1 | 20 (25%) | 85 (44%) | 55 (30%) |
| Grade 2 | 20 (25%) | 59 (31%) | 59 (32%) |
| Grade 3 | 39 (49%) | 54 (28%) | 68 (37%) |
| Girls | 45 (57%) | 93 (48%) | 82 (45%) |
| Age (years) | 17.86 (SD = 1.09) | 17.3 (SD = 1.10) | 17.53 (SD = 1.14) |

Note: % from all the students in this condition.

question, "How important is it for you to access credible news?" rated on a five-point scale from *not at all important* (1) to *extremely important* (5). Participants' confidence in navigating digital news was measured using the question, "How easy do you find it to locate the information you need on the internet?" with responses on a five-point scale ranging from *not at all easy* (1) to *extremely easy* (5). The perceived importance of democracy was assessed with the question from [51], "How important is it for you to live in a country governed democratically?" rated on a ten-point scale from *not at all important* (1) to *extremely important* (10).

To assess actively open-minded thinking (AOT), we employed a 10-item version of the scale [57]. However, we were unable to replicate the original factor structure. Following an exploratory factor analysis (see Table C1 in Appendix C), we selected a shortened version consisting of five reverse-scored items: "I tend to view people as either for me or against me", "It is more important to be loyal to one's principles and ideals than to be open", "I believe there are many wrong ways, but only one right answer in almost all questions", "I believe it only confuses and misleads students to let them listen to controversial speakers", "One should ignore evidence that conflicts with one's fundamental beliefs". The final scale demonstrated acceptable internal consistency (Cronbach's $\alpha = 0.74$). Table 2 provides an overview of the four analyses and the variables used in them. See Table D1 in Appendix D for the summary of all variables.

We used a selection of test items and shortened scales to avoid fatigue among the participants. Still, we did get comments from some participants stating that the questionnaire was too long.

## Analysis

The sample in this study comes from just one school, but students are grouped into classes and two observation points. Given this structure, similar multilevel models were used across the four analyses, with observations grouped for students (pretest and posttest for each student) and students grouped in classes (study group). Intervention effects were estimated as an interaction term between treatment and time, both coded as dummy variables. As shown by [58], this model specification (three levels and an interaction term) gives estimates similar to alternative estimation approaches for the same study design. Linear models were used for reliability ratings, and logistic models were used for digital tools. The analysis was done in R using the lmer4 package [59].

For the analysis of the first two interventions (Bad News and News Evaluator), each intervention group was compared to the control group after the second intervention group was excluded. This was done to simplify the presentation of results and does not change the estimates. The resulting number of students for the analysis of Bad News and News

**Table 2. Overview of analyses and variables.**

| Analyses | Independent variable | Dependent variable |
|---|---|---|
| 1<br>Inoculation intervention | Participation in Bad News game | Reliability ratings of misleading and credible tweets from one (rated as very likely false) to seven (rated as very likely true), four items separately, |
| 2<br>Fact-checking skills intervention | Participation in News Evaluator workshop | Reliability ratings of misleading and credible Facebook posts from one to seven, two items |
| | | Use of digital tools when doing the test items (binary) |
| 3<br>Disciplinary literacy interventions | Participation in subject-specific teaching about misinformation (Social sciences, History, Psychology, Natural Sciences, Art) | Mean reliability ratings for true news and false news, from one to seven |
| 4<br>Factors of identifying true and false news | Importance of getting credible news (0–6) | Mean reliability ratings for true news and false news, from one to seven |
| | Self-rated confidence in navigating digital news (1–5) | |
| | Importance of living in a democratic country (1–10) | |
| | Shortened version of AOT (0–30) | |
| | Vocational track | |

Evaluator is 272 and 260, respectively. As shown by Tables 4 and 6 (see coefficient for the experimental group, "Bad News" and "News Evaluator", respectively), there are no significant differences between the control and experimental groups in the pretest.

For the subject-specific interventions, students who did not participate in any intervention (N = 204) were used as a control group for all five interventions. An alternative approach would be to have a different control group for every subject intervention and to each time compare those who participated in this specific intervention and those who did not, even if they received instruction in some other subject. This approach gives the same results, so the first was chosen for simplicity (Table E1 in Appendix E). In addition to estimating the effects of each subject intervention separately, we checked if participating in any of them has an effect if taken together.

A complete case analysis was performed to deal with missing data. The missing data rate was higher for the tweet items (9%, 13%, 13%, and 11%), while 1% for the rest of the test items.

Data and R code are available in S1 File.

## Results

### Inoculation intervention (Bad News)

Table 3 shows the mean and standard deviations for reliability ratings of the four tweets for students who did and did not play the Bad News game. Table 4 reports the results of the regressions used to estimate the effect of this intervention. It shows that the interaction term coefficient between treatment (Bad News) and time (Posttest) is insignificant for all four items, implying a lack of long-term effects.

### Fact-checking skills intervention (News evaluator)

Table 5 shows the mean and standard deviations for reliability ratings of the two Facebook posts and the use of digital tools for students who did and did not take part in the News Evaluator workshop. Table 6 reports the results of the regressions used to estimate the effect of this intervention (linear for posts and logistic for digital tools). It shows that the coefficient of the interaction term between treatment and time is insignificant for all the outcomes, which implies the lack of long-term effects. However, there might be some indication of a positive effect for the first Facebook post ($p = 0.056$, see Table F1 in Appendix F). Thus, students participating in the News Evaluator workshop may have become better than students in the control group at fact-checking a reliable news post.

As a fact-checking skills intervention, the News Evaluator workshop assumes that students become better at identifying true and false news because they are taught to use digital tools. Fig 1 shows the mean reliability judgments of students who did and did not use digital tools across the tests and intervention conditions. Again, it provides no support for the expected effect of the intervention.

### Disciplinary literacy interventions (five subjects)

Table 7 shows mean and standard deviations for mean reliability ratings of true and false news for students who did and did not participate in subject-specific interventions. Unlike previous analyses, intervention conditions here are not exclusive, as each class could participate in up to three subject-specific interventions. Table 8 reports the results of the regressions used to estimate the effect of this intervention. It shows that the coefficient of the interaction term between treatment and time is insignificant for both true and false news, which implies the lack of long-term effect.

### Factors of identifying true and false news

Table 9 provides regression results for the factors of identifying true and false news. It suggests that the effects of these factors differ for true and false news. Credibility importance has a highly significant positive effect on the mean reliability

**Table 3. Means and standard deviations for reliability ratings of tweets, Bad News intervention.**

|  |  | Tweet 1 (false) | | Tweet 2 (false) | | Tweet 3 (false) | | Tweet 4 (true) | |
|---|---|---|---|---|---|---|---|---|---|
|  |  | Mean | SD | Mean | SD | Mean | SD | Mean | SD |
| Control | Pretest | 2.84 | 1.94 | 2.75 | 1.75 | 2.56 | 1.37 | 4.77 | 2.09 |
|  | Posttest | 3.13 | 2.06 | 2.61 | 1.51 | 2.69 | 1.31 | 4.99 | 2.01 |
| Bad News | Pretest | 2.90 | 2.05 | 2.73 | 1.64 | 2.56 | 1.51 | 4.96 | 1.88 |
|  | Posttest | 2.86 | 2.09 | 2.90 | 1.73 | 2.71 | 1.54 | 5.08 | 1.97 |

**Table 4. Standardised regression coefficients and standard errors for reliability ratings of tweets, Bad News intervention.**

|  | Tweet 1 (false) | Tweet 2 (false) | Tweet 3 (false) | Tweet 4 (true) |
|---|---|---|---|---|
| Intercept | −0.09 (0.13) | −0.04 (0.14) | −0.11 (0.13) | −0.19 (0.16) |
| Posttest | 0.16 (0.13) | −0.10 (0.16) | 0.08 (0.14) | 0.11 (0.12) |
| Bad News | 0.01 (0.15) | −0.06 (0.16) | −0.04 (0.15) | 0.11 (0.19) |
| Posttest x Bad News | −0.18 (0.15) | 0.20 (0.19) | 0.02 (0.16) | −0.04 (0.14) |
| N students | 273 | 272 | 270 | 268 |
| N classes | 24 | 24 | 24 | 24 |

Note:

$^*p < 0,05$,

$^{**}p < 0,01$,

$^{***}p < 0,001$.

**Table 5. Means and standard deviations for reliability ratings of Facebook posts and use of digital tools, News Evaluator intervention.**

|  |  | Post 1 (true) | | Post 2 (false) | | Use of digital tools | |
|---|---|---|---|---|---|---|---|
|  |  | Mean | SD | Mean | SD | Mean | SD |
| Control | Pretest | 2.54 | 1.79 | 1.55 | 3.46 | 0.14 | 0.35 |
|  | Posttest | 2.58 | 1.66 | 1.26 | 3.68 | 0.15 | 0.36 |
| News Evaluator | Pretest | 2.43 | 1.72 | 1.32 | 3.72 | 0.12 | 0.32 |
|  | Posttest | 2.94 | 1.50 | 1.58 | 3.83 | 0.13 | 0.34 |

**Table 6. Standardised regression coefficients and standard errors for reliability ratings of Facebook posts and use of digital tools, News Evaluator intervention.**

|  | Post 1 (true) | Post 2 (false) | Use of digital tools |
|---|---|---|---|
| Intercept | −0.05 (0.12) | −0.16 (0.11) | −1.98$^{***}$ (0.50) |
| Posttest | 0.04 (0.13) | 0.13 (0.14) | 0.08 (0.49) |
| News Evaluator | −0.08 (0.15) | 0.16 (0.14) | −0.45 (0.56) |
| Posttest x News Evaluator | 0.31 (0.16) | −0.07 (0.17) | 0.11 (0.60) |
| N students | 260 | 260 | 260 |
| N classes | 28 | 28 | 28 |

Note:

$^*p < 0,05$,

$^{**}p < 0,01$,

$^{***}p < 0,001$.

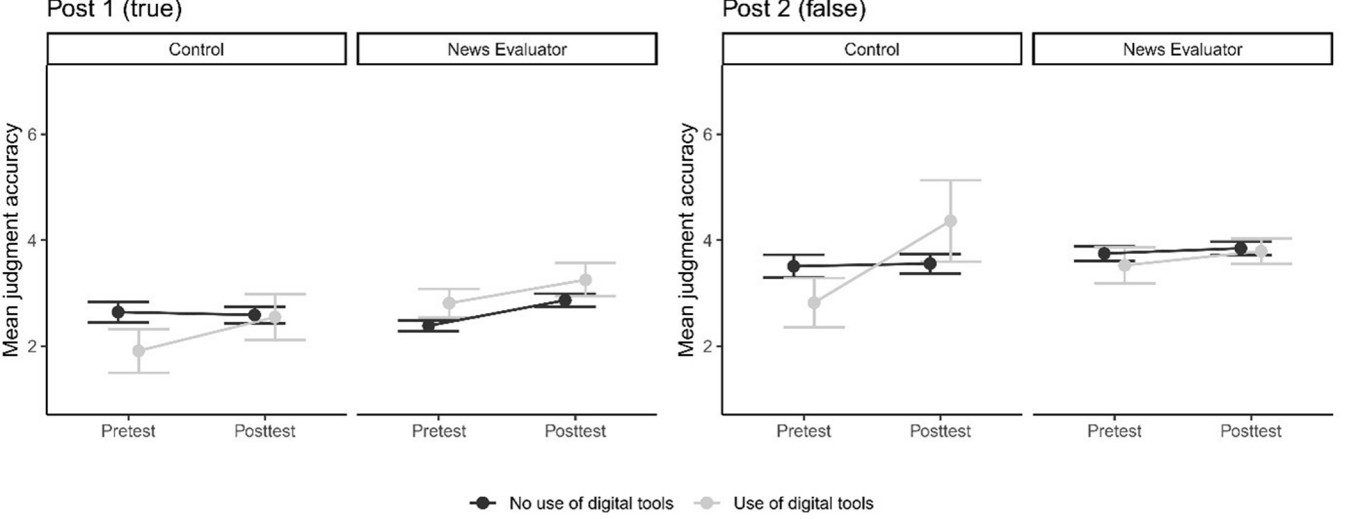

**Fig 1. Mean reliability judgment for Facebook posts by intervention group and the use of digital tools (with standard errors).**

Table 7. Means and standard deviations for reliability ratings of true and false news, subject-specific interventions.

| Intervention | N of classes | N of students | Test | True news | | False news | |
|---|---|---|---|---|---|---|---|
| | | | | Mean | SD | Mean | SD |
| None | 22 | 204 | Pretest | 3.38 | 0.98 | 2.94 | 0.97 |
| | | | Posttest | 3.61 | 1.08 | 3.09 | 1.01 |
| Any of the five subjects | 21 | 255 | Pretest | 3.62 | 0.84 | 3.04 | 0.87 |
| | | | Posttest | 3.88 | 0.87 | 3.18 | 0.89 |
| Social sciences | 9 | 145 | Pretest | 3.63 | 0.83 | 3.12 | 0.89 |
| | | | Posttest | 3.91 | 0.85 | 3.19 | 0.95 |
| History | 11 | 111 | Pretest | 3.61 | 0.84 | 3.04 | 0.92 |
| | | | Posttest | 3.88 | 0.94 | 3.30 | 0.98 |
| Psychology | 3 | 57 | Pretest | 3.62 | 0.93 | 2.98 | 0.79 |
| | | | Posttest | 3.79 | 0.82 | 3.11 | 0.94 |
| Natural sciences | 4 | 18 | Pretest | 3.37 | 0.78 | 2.94 | 1.02 |
| | | | Posttest | 3.78 | 0.80 | 3.23 | 0.82 |
| Art | 2 | 21 | Pretest | 3.49 | 0.81 | 3.02 | 0.56 |
| | | | Posttest | 3.52 | 0.86 | 3.14 | 0.79 |

rating of true news, as expected, but no effect on false news. Similarly, and in line with expectations, the vocational track is a negative factor in identifying true news but has no effect on false news. In contrast, the importance of democracy shows the expected effect on the mean reliability rating of false news, but there is none for true news. Noteworthy, AOT has a significant negative effect on mean reliability ratings of both true and false news. In other words, students high in AOT in this sample are more sceptical towards all news, regardless of whether they are true or false. This underscores the importance of distinguishing between appropriate scepticism, questioning unverified claims, and indiscriminate doubt, which may lead to the rejection of credible information. Understanding this distinction is vital in interpreting how AOT functions in misinformation detection. Confidence in one's own source-critical abilities has no significant effect on either type of news.

**Table 8. Estimated regression coefficients and standard errors for mean reliability ratings of true and false news, subject-specific interventions.**

| Intervention | | True news | False news |
|---|---|---|---|
| Any | Intercept | −0.29** (0.09) | −0.14 (0.07) |
| | Posttest | 0.24** (0.08) | 0.16* (0.07) |
| | Intervention | 0.26* (0.12) | 0.11 (0.1) |
| | Posttest x Intervention | 0.04 (0.1) | −0.01 (0.1) |
| | N students | 458 | 458 |
| | N classes | 43 | 43 |
| Social sciences | Intercept | −0.25** (0.09) | −0.14 (0.08) |
| | Posttest | 0.23** (0.08) | 0.15* (0.07) |
| | Intervention | 0.28 (0.15) | 0.19 (0.12) |
| | Posttest x Intervention | 0.05 (0.12) | −0.08 (0.11) |
| | N students | 348 | 348 |
| | N classes | 31 | 31 |
| History | Intercept | −0.22* (0.09) | −0.14 (0.08) |
| | Posttest | 0.23** (0.08) | 0.15* (0.07) |
| | Intervention | 0.23 (0.16) | 0.11 (0.13) |
| | Posttest x Intervention | 0.04 (0.13) | 0.12 (0.12) |
| | N students | 314 | 314 |
| | N classes | 33 | 33 |
| Psychology | Intercept | −0.17 (0.09) | −0.09 (0.08) |
| | Posttest | 0.23** (0.08) | 0.15* (0.07) |
| | Intervention | 0.24 (0.22) | 0.05 (0.2) |
| | Posttest x Intervention | −0.06 (0.16) | −0.01 (0.15) |
| | N students | 260 | 260 |
| | N classes | 25 | 25 |
| Natural sciences | Intercept | −0.13 (0.09) | −0.09 (0.08) |
| | Posttest | 0.22** (0.08) | 0.15* (0.07) |
| | Intervention | 0 (0.28) | 0.03 (0.27) |
| | Posttest x Intervention | 0.18 (0.27) | 0.15 (0.24) |
| | N students | 221 | 221 |
| | N classes | 26 | 26 |
| Art | Intercept | −0.13 (0.09) | −0.09 (0.08) |
| | Posttest | 0.22** (0.08) | 0.15* (0.07) |
| | Intervention | 0.12 (0.3) | 0.09 (0.28) |
| | Posttest x Intervention | −0.19 (0.26) | −0.03 (0.23) |
| | N students | 224 | 224 |
| | N classes | 24 | 24 |

Note:

*$p < 0,05$,

**$p < 0,01$,

***$p < 0,001$.

**Table 9. Estimated regression coefficients and standard errors for mean reliability ratings of true and false news, and factors of identifying true and false news.**

|  | True news | False news |
|---|---|---|
| Intercept | 0.02 (0.07) | −0.06 (0.07) |
| Posttest | 0.24*** (0.05) | 0.13** (0.05) |
| Credibility importance | 0.12*** (0.03) | 0.05 (0.03) |
| Confidence | 0.02 (0.03) | −0.04 (0.03) |
| Importance of democracy | 0.05 (0.03) | −0.1** (0.03) |
| AOT | −0.15*** (0.04) | −0.21*** (0.04) |
| Vocational track | −0.34*** (0.09) | −0.05 (0.09) |

Note: All continuous variables are standardised.

*$p < 0,05$,

**$p < 0,01$,

***$p < 0,001$. N (students) = 457, N (classes) = 43.

## Conclusion

This study investigated the long-term effects of interventions aimed at improving upper-secondary students' ability to evaluate digital news and misinformation critically. In response to RQ1, the study found no long-term effects of the interventions. The News Evaluator workshop showed only a marginal improvement on one item, but neither the Bad News game nor the disciplinary literacy interventions led to sustained improvements in students' ability to evaluate true and false news at the three-month follow-up. Regarding RQ2, several individual-level factors were associated with students' performance. Students who valued credibility and democratic ideals performed better in evaluating news, while higher AOT scores were linked to generalised scepticism toward both true and false news. Self-rated confidence did not predict actual ability, and students in vocational tracks performed worse than those in theoretical programs.

Specifically, we tested the impact of inoculation against misinformation through the *Bad News* game (H1), fact-checking skills via the *News Evaluator* workshop (H2, H3), and subject-specific disciplinary literacy interventions. Additionally, we examined factors related to students' abilities to evaluate digital news, including credibility importance (H4), the importance of democracy (H5), actively open-minded thinking (H6), self-rated confidence (H7), and vocational track status (H8).

Contrary to H1 and H2, neither the *Bad News* game nor the *News Evaluator* workshop significantly affected students' ability to distinguish between misleading and credible news. This aligns with concerns raised in previous research that the effects of prebunking and fact-checking interventions often diminish over time without reinforcement [12]. Although the *News Evaluator* workshop group showed an improvement for one item ($p = 0.056$), the impact was not as significant as expected. Students in this intervention group also did not use digital verification tools more frequently three months after the workshop (H3), indicating that a short and scalable fact-checking workshop alone may not be enough to create lasting changes in behaviour.

Similarly, despite being tailored to different disciplines, the subject-specific interventions did not produce significant long-term improvements in students' abilities to evaluate true and false news. These findings suggest that while integrating misinformation education into subject teaching holds promise, standalone interventions may be insufficient for building durable source evaluation skills. Given that prior research highlights the importance of subject-specific knowledge when evaluating digital information [10,11], future research should explore whether more sustained and embedded disciplinary approaches yield better results. It is noteworthy that students in the control group from somewhat higher grades and theoretical programs did not significantly outperform their peers in pretest evaluations. This aligns with earlier findings showing that educational orientation alone does not guarantee proficiency in digital civic literacy (10).

As expected, students who rated access to credible information as important (H4) were significantly better at identifying accurate news, reinforcing previous research showing that valuing credibility is crucial for resisting misinformation [10,11,44]. However, in contrast to this previous research, recognising the importance of access to credible news was not associated with a better ability to evaluate misleading news. In line with H5, we found that students who rated the importance of democracy higher (H5) were better at identifying false news, aligning with previous studies linking democratic ideals to abilities to evaluate (mis)information [51].

Our findings on actively open-minded thinking (AOT) (H6) diverge from prior research. While previous studies indicate that AOT is positively associated with better discernment between true and false news [52], our results suggest something different. In this study, higher AOT scores were correlated with increased scepticism toward both credible and false news.

Additionally, self-rated confidence in online fact-checking (H7) did not predict students' ability to evaluate digital news, reinforcing earlier research indicating that subjective confidence is often a poor indicator of media and information literacy [10,11]. Finally, as expected, students in vocational tracks (H8) performed worse in identifying true news, consistent with findings that educational background significantly influences students' digital civic literacy [10].

## Discussion

While prior research has shown short-term impacts on skills linked to inoculation against misinformation ($d = 0.3$; [4]) and fact-checking ($d = 0.65$; [5]) in classroom settings, our findings indicate that these effects do not necessarily persist over time. The lack of long-term impact from the *Bad News* game and *News Evaluator* workshop suggests that without reinforcement, students may revert to pre-intervention behaviours, highlighting the need for repeated exposure and including, for instance, retrieval practice to consolidate learning [12,60].

Previous research has noted the lingering prebunking effect from playing the *Bad News* game [12,54], and here we see that this is true also in school settings where students may be exposed to assignments stimulating critical thinking across multiple subjects. Apparently, this did not work as "booster shots" for the "inoculated" students against misinformation. This study also shows a similar lingering effect after interventions designed to support fact-checking skills by using lateral reading with a stronger short-term impact. One key limitation of this intervention appears to be the inability to instil habits of using digital verification tools. The fact that students in the *News Evaluator* workshop did not use digital verification tools more frequently three months after the intervention suggests that even when students are trained in fact-checking strategies, they may not consistently apply these skills over time and in their everyday media consumption. This adds a dimension to previous findings that have shown an increase in lateral reading behaviours immediately following such training [5,30]. The results highlight the challenge of translating learning fact-checking in schools into habitual practice and supporting students' technocognition, emphasising the importance of designing interventions that encourage continued engagement beyond the classroom.

Students in the *News Evaluator* workshop did become somewhat better at evaluating a news post that is difficult to fact-check, showing some long-term promise in this approach. However, the very low percentages of people using digital tools in both intervention and control groups show a significant challenge regarding young people's technocognition. Knowing when and how to use technology to support cognition when evaluating information seems very limited. The low percentages presented in Table 5 echo a national survey with 4157 adults and university students, where only 15% used digital tools to evaluate the same test items that we used here [61]. The short-term effect of the workshop is usually that more than half of the participants would start using digital tools to fact-check (mis)information [5,30]. Thus, while students may acquire fact-checking abilities in controlled educational settings, no more than 15% reported using digital tools to support their fact-checking in the posttest, even after being trained in lateral reading strategies. This suggests that technocognition [25] remains limited over time among students despite the targeted interventions.

Roozenbeek et al. [3] have called for more research on misinformation interventions in ordinary school settings, emphasising the importance of understanding how these strategies function in real educational environments. However,

they also note that achieving significant impact in real-world settings can be more challenging than in lab environments. The present study adds to this concern, as the interventions that have shown promise in prior experimental classroom research did not translate into substantial long-term improvements in identifying misinformation and fact-checking skills. Future research should explore how to bridge this gap, examining ways to enhance the ecological validity and effectiveness of classroom-based misinformation education.

While designed to leverage disciplinary literacy focused on critical thinking, the subject-specific interventions failed to show significant long-term effects. One possible explanation is that short, standalone interventions may be insufficient to strengthen the deep knowledge structures necessary for effective disciplinary thinking [42]. Unlike media and information literacy training, which may focus on heuristic strategies such as lateral reading, disciplinary literacy may depend on sustained engagement with content-specific knowledge. The findings suggest that future interventions should be embedded within broader curricular frameworks rather than delivered as isolated workshops.

Prior research has linked AOT to better misinformation detection, as it is associated with reduced fact resistance, low dogmatism and an increased willingness to update beliefs in light of new evidence [52,61]. However, in this study, higher AOT scores correlated with scepticism toward both true and false news. This suggests that instead of enhancing discernment, AOT may in this context reflect a broader tendency toward generalised distrust. It is possible that the shortened AOT instrument used here primarily captures this sceptical stance rather than thoughtful openness. The Swedish school context and adolescent age group may also shape how AOT is expressed and measured, raising questions about its developmental validity. Future research should examine whether different dimensions of AOT—such as analytical thinking versus general scepticism—play distinct roles in misinformation resistance and if this may differ between teenagers and older people paid to answer online surveys.

As expected, students who valued credibility were better at identifying true news, which aligns with research showing that credibility importance is a key factor in misinformation resistance [10,11,44]. Similarly, students who placed a high value on democracy were better at identifying false news, supporting previous findings linking democratic ideals to media literacy skills [51,61].

Our findings also underscore existing disparities in digital civic literacy. Students in vocational tracks performed worse in identifying trustworthy news, mirroring previous studies highlighting differences in source evaluation abilities based on educational background [10]. This suggests that interventions should be tailored to address these gaps, ensuring that misinformation education is accessible and effective across diverse student populations. Additionally, the lack of correlation between self-rated skills and abilities to evaluate news further reinforces the idea that confidence in one's digital literacy skills does not necessarily translate to better discernment [11].

In light of these results, future misinformation interventions should explore new approaches to strengthening long-term effects. Research suggests that repeated exposure, spaced retrieval, and real-world application exercises may help reinforce learning over time [60,62]. Additionally, integrating AI-driven fact-checking tools and interactive misinformation simulations into students' everyday digital environments may provide opportunities and challenges for sustained engagement beyond traditional classroom instruction.

Taken together, these findings highlight both the promise and the limitations of current misinformation education strategies. While prebunking, fact-checking, and disciplinary literacy interventions offer valuable tools for improving students' digital source criticism, their long-term effectiveness remains uncertain. Addressing these challenges will require a shift toward more sustained and integrated approaches that not only teach critical thinking but also foster habitual engagement with credible information sources.

## Limitations

Several limitations should be noted when interpreting these findings. The study was conducted within a single school, limiting the results' generalisability to broader populations. While the sample included students from different educational

tracks, replication in diverse educational settings is needed to determine whether similar patterns hold across different demographic and institutional contexts.

Moreover, the study relied on a quasi-experimental design, meaning that a full randomisation of participants into intervention groups was not possible. Although efforts were made to ensure a balanced distribution of students across conditions, some differences in baseline characteristics may have influenced the results.

Questionnaire fatigue may have affected students' engagement and response accuracy, particularly in the posttest and later sections of the assessment. As attention and motivation declined, responses may have become less thoughtful or inconsistent, especially for tasks requiring nuanced judgment, such as identifying and evaluating true and false news. This could have weakened the observed relationships between intervention effects and misinformation discernment. Future research should address this by shortening assessments, incorporating interactive elements, or using adaptive testing to maintain engagement and data reliability.

In addition, the measurement instruments used to assess misinformation detection were adapted from previous studies; however, some exhibited relatively low internal consistency. This may have introduced variability in the results, particularly for the subject-specific interventions. The shortened AOT scale may not have fully captured the construct of actively open-minded thinking in this adolescent school sample. The relationship between scepticism and critical thinking likely varies by age and context, highlighting the need for age-appropriate measures when evaluating such constructs in educational settings. Future research should refine these measures to ensure greater reliability and validity in assessing students' ability to evaluate digital news.

Finally, the study was conducted in a school setting, with participants being aware that they were part of a study with a focus on critical thinking in a digital world. Students' behaviour in this setting will not fully capture how they engage with news in real-world settings. Incorporating behavioural tracking methods, such as monitoring students' online verification behaviours over time, could provide a more accurate picture of how misinformation interventions may influence real-world decision-making.

## Future research

The absence of long-term effects from the interventions highlights key challenges in designing effective and enduring media and information literacy programs. Previous research suggests that boosts, spaced repetition and retrieval practice—educational strategies shown to enhance long-term retention—may be necessary to sustain the benefits of prebunking and fact-checking training [12,60,62]. Future studies should explore whether periodic reinforcement sessions or adaptive learning models can improve the longevity of classroom intervention with positive short-term effects.

Moreover, while the subject-specific interventions did not yield significant effects, this does not necessarily indicate that disciplinary literacy is ineffective. Instead, the findings suggest that short-term disciplinary approaches may be insufficient, and more integrated, long-term strategies may be required. Prior research on disciplinary literacy emphasises that subject knowledge is linked to critical thinking [42,43], highlighting the need for interventions that incorporate source criticism more systematically across different subjects.

The results on AOT in relation to false news raise critical questions about how to balance scepticism and trust in misinformation education. Future studies should examine whether different aspects of AOT, such as analytical thinking versus scepticism, play distinct roles in misinformation resistance and how interventions can be designed to encourage critical engagement without promoting indiscriminate doubt.

Moreover, the transfer issue extends beyond the immediate application of fact-checking skills to a broader challenge of preparing students for future misinformation environments. Digital media landscapes evolve rapidly, with new forms of misinformation emerging continuously [63]. An intervention that is effective today may be less relevant in the future if students are not equipped with adaptable strategies for navigating changing information ecosystems. This raises questions about how misinformation education can be designed to foster resilience that persists across different platforms, media formats, and societal contexts.

Given the increasing prevalence of algorithm-driven misinformation, future interventions should consider integrating digital tools and AI-based verification strategies into educational programs. While traditional classroom-based approaches remain valuable, scalable digital interventions, such as AI fact-checking assistants or misinformation simulations embedded in students' real-world media environments, may offer more effective ways to foster long-term media and information literacy. Such studies should include different types of boosters and measurements to investigate long-term effects in nuanced ways.

In conclusion, this study contributes to the field of research on misinformation education by highlighting the challenges of achieving long-term effects through short-term interventions. The findings underscore the need for more sustained and embedded approaches that integrate cognitive science principles, disciplinary literacy, and digital tools to support students in critically engaging with online information.

## Appendix A

### Power analysis

**Table A1. Power estimation for pairwise comparisons using PowerUP! tool (2-level cluster-randomised assignment, CRA2_2).**

| Assumptions | | Comments |
|---|---|---|
| Alpha Level ($\alpha$) | 0,05 | Probability of a Type I error |
| Two-tailed or One-tailed Test? | 2 | |
| Power (1-$\beta$) | 0,80 | Statistical power (1-probability of a Type II error) |
| Rho (ICC) | 0,05 | Proportion of variance in outcome that is between clusters |
| P | 0,66 | Proportion of Level 2 units randomised to treatment: $J_T / (J_T + J_C)$ |
| $R_1^2$ | 0,20 | Proportion of variance in Level 1 outcomes explained by Level 1 covariates |
| $R_2^2$ | 0,00 | Proportion of variance in Level 2 outcome explained by Level 2 covariates |
| g* | 0 | Number of Level 2 covariates |
| n (Average Cluster Size) | 20 | Mean number of Level 1 units per Level 2 cluster (harmonic mean recommended) |
| J (Sample Size [# of Clusters]) | 30 | Number of Level 2 units |
| M (Multiplier) | 2,90 | Computed from $T_1$ and $T_2$ |
| $T_1$ (Precision) | 2,05 | Determined from alpha level, given two-tailed or one-tailed test |
| $T_2$ (Power) | 0,85 | Determined from the given power level |
| **MDES** | **0,332** | **Minimum Detectable Effect Size** |

## Appendix B

### Sample for subject-specific interventions

**Table B1. Characteristics of the sample for subject-specific interventions.**

| Group | N st | N cl | Vocational track | | Grade 1 | | Grade 2 | | Grade 3 | | Girls | | Age | |
|---|---|---|---|---|---|---|---|---|---|---|---|---|---|---|
| | | | N | % | N | % | N | % | N | % | N | % | M | SD |
| Control | 204 | 22 | 150 | 74% | 93 | 46% | 67 | 33% | 44 | 22% | 78 | 38% | 17.33 | 1.17 |
| Any | 255 | 21 | 61 | 24% | 67 | 26% | 71 | 28% | 117 | 46% | 142 | 56% | 17.61 | 1.08 |
| Social sciences | 145 | 9 | 16 | 11% | 67 | 46% | 44 | 30% | 34 | 23% | 86 | 59% | 17.15 | 1.02 |
| History | 111 | 11 | 38 | 34% | 61 | 55% | 27 | 24% | 23 | 21% | 47 | 42% | 17.03 | 1.00 |
| Psychology | 57 | 3 | 0 | 0% | 15 | 26% | 0 | 0% | 42 | 74% | 42 | 74% | 17.91 | 1.14 |
| Natural sciences | 18 | 4 | 18 | 100% | 0 | 0% | 0 | 0% | 18 | 100% | 8 | 44% | 18.56 | 0.62 |
| Arts | 21 | 2 | 0 | 0% | 0 | 0% | 11 | 52% | 10 | 48% | 18 | 86% | 18.14 | 1.06 |

## Appendix C

## Measurements

**Table C1. Factor loadings for 2-factor EFA solution for AOT (only > 0.5).**

| AOT item | Factor1 | Factor2 |
|---|---|---|
| aot4 | 0.53 | |
| aot6 | 0.61 | |
| aot7 | 0.59 | |
| aot8 | 0.67 | |
| aot9 | 0.76 | |
| aot2 | | 0.85 |
| aot3 | | 0.80 |
| aot5 | | 0.53 |
| aot1 | | |
| aot10 | | |

Variance explained by factor 1= 0.23, factor 2 = 0.19. Correlation between factors = -0.44.

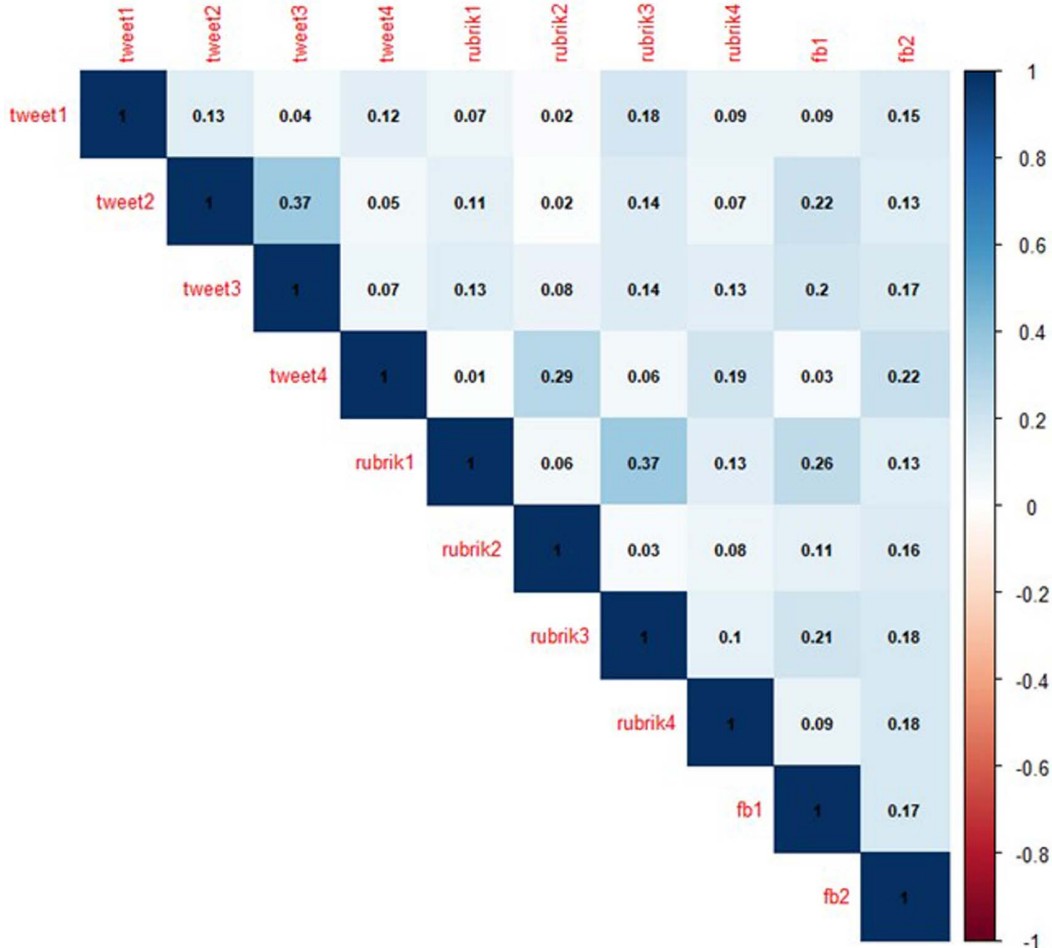

Fig C1. **Correlation between all test items.**

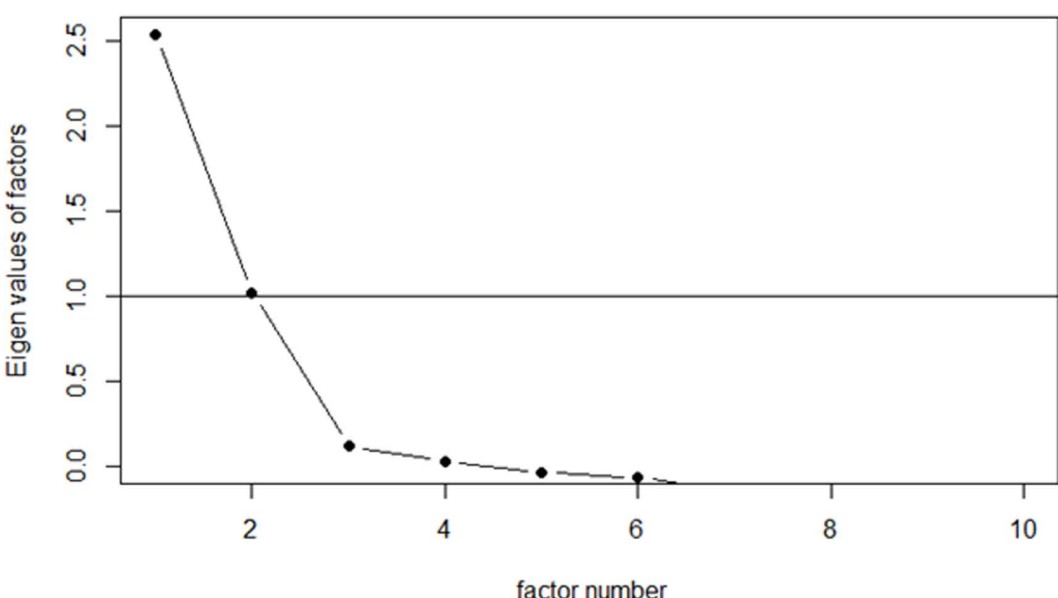

Fig C2. **Scree plot for AOT scale (10 items).**

## Appendix D

## Variables summary

**Table D1.  Variables summary.**

| Variable | Mean | SD | Median | Min | Max | N |
|---|---|---|---|---|---|---|
| Tweet 1 (false) | 3.05 | 2.07 | 2 | 1 | 7 | 835 |
| Tweet 2 (false) | 2.85 | 1.61 | 3 | 1 | 7 | 797 |
| Tweet 3 (false) | 2.76 | 1.49 | 2 | 1 | 7 | 796 |
| Tweet 4 (false) | 4.98 | 1.9 | 5 | 1 | 7 | 813 |
| Post 1 (true) | 2.63 | 1.45 | 2 | 1 | 7 | 907 |
| Post 2 (false) | 3.59 | 1.66 | 4 | 1 | 7 | 907 |
| Use of digital tools | 0.12 | 0.33 | 0 | 0 | 1 | 901 |
| Credibility importance | 3.7 | 1.14 | 4 | 0 | 6 | 918 |
| Confidence | 3.93 | 0.82 | 4 | 1 | 5 | 918 |
| Importance of democracy | 8.32 | 2.14 | 9 | 1 | 10 | 894 |
| AOT | 15.64 | 4.87 | 15 | 0 | 30 | 918 |
| True news | 3.64 | 0.95 | 3.75 | 1 | 7 | 909 |
| False news | 3.07 | 0.93 | 3 | 1 | 7 | 911 |
| Girl | 0.48 | 0.5 | 0 | 0 | 1 | 918 |
| Age | 17.49 | 1.13 | 17 | 16 | 22 | 914 |
| Vocational track | 0.46 | 0.5 | 0 | 0 | 1 | 918 |

## Appendix E

### Alternative estimations

**Table E1. Regression results for inoculation intervention (Bad News), full sample and the Bad News group as reference (standardised).**

| Outcome | Coefficient | Estimate | SE | P value |
|---|---|---|---|---|
| Tweet 1 (false) | Intercept | −0.08 | 0.08 | 0.349 |
| | Posttest | −0.02 | 0.08 | 0.812 |
| | Control | −0.01 | 0.15 | 0.945 |
| | News Evaluator | 0.11 | 0.12 | 0.362 |
| | Posttest x Control | 0.18 | 0.15 | 0.227 |
| | Posttest x News Evaluator | 0.18 | 0.11 | 0.111 |
| Tweet 2 (false) | Intercept | −0.10 | 0.09 | 0.246 |
| | Posttest | 0.10 | 0.09 | 0.295 |
| | Control | 0.07 | 0.16 | 0.673 |
| | News Evaluator | 0.19 | 0.13 | 0.148 |
| | Posttest x Control | −0.21 | 0.17 | 0.240 |
| | Posttest x News Evaluator | −0.06 | 0.14 | 0.656 |
| Tweet 3 (false) | Intercept | −0.15 | 0.08 | 0.077 |
| | Posttest | 0.10 | 0.09 | 0.258 |
| | Control | 0.04 | 0.15 | 0.815 |
| | News Evaluator | 0.28 | 0.12 | 0.020 |
| | Posttest x Control | −0.02 | 0.17 | 0.897 |
| | Posttest x News Evaluator | −0.09 | 0.13 | 0.476 |
| Tweet 4 (true) | Intercept | −0.07 | 0.10 | 0.470 |
| | Posttest | 0.06 | 0.08 | 0.427 |
| | Control | −0.11 | 0.18 | 0.542 |
| | News Evaluator | −0.04 | 0.14 | 0.805 |
| | Posttest x Control | 0.04 | 0.15 | 0.768 |
| | Posttest x News Evaluator | 0.10 | 0.11 | 0.396 |

**Table E2. Regression results for fact-checking skills intervention (News Evaluator), full sample and the News Evaluator group as reference (standardised).**

| Outcome | Coefficient | Estimate | SE | P value |
|---|---|---|---|---|
| Post 1 (true) | Intercept | −0,14 | 0,07 | 0,064 |
| | Posttest | 0,35 | 0,09 | 0,000 |
| | Control | 0,08 | 0,13 | 0,557 |
| | Bad News | −0,05 | 0,10 | 0,648 |
| | Posttest x Control | −0,31 | 0,16 | 0,052 |
| | Posttest x Bad News | −0,02 | 0,12 | 0,847 |
| Post 2 (false) | Intercept | 0,08 | 0,08 | 0,339 |
| | Posttest | 0,06 | 0,09 | 0,477 |
| | Control | −0,15 | 0,15 | 0,292 |
| | Bad News | −0,31 | 0,11 | 0,008 |
| | Posttest x Control | 0,07 | 0,16 | 0,678 |
| | Posttest x Bad News | 0,16 | 0,12 | 0,207 |

| Outcome | Coefficient | Estimate | SE | P value |
|---|---|---|---|---|
| Use of digital tools | Intercept | −0,15 | 0,08 | 0,077 |
| | Posttest | 0,10 | 0,09 | 0,258 |
| | Control | 0,04 | 0,15 | 0,815 |
| | Bad News | 0,28 | 0,12 | 0,020 |
| | Posttest x Control | −0,02 | 0,17 | 0,897 |
| | Posttest x Bad News | −0,09 | 0,13 | 0,476 |

**Table E3. Regression results for subject-specific interventions, using the rest as control (standardised).**

| Intervention | Outcome | Coefficient | Estimate | SE | P value |
|---|---|---|---|---|---|
| Social sciences | True news | Intercept | −0,21 | 0,07 | 0,006 |
| | | Posttest | 0,25 | 0,06 | 0,000 |
| | | Intervention group | 0,21 | 0,14 | 0,144 |
| | | Posttest x Intervention group | 0,04 | 0,11 | 0,735 |
| | False news | Intercept | −0,14 | 0,06 | 0,024 |
| | | Posttest | 0,19 | 0,06 | 0,002 |
| | | Intervention group | 0,19 | 0,10 | 0,083 |
| | | Posttest x Intervention group | −0,11 | 0,11 | 0,288 |
| History | True news | Intercept | −0,19 | 0,07 | 0,010 |
| | | Posttest | 0,25 | 0,06 | 0,000 |
| | | Intervention group | 0,15 | 0,15 | 0,315 |
| | | Posttest x Intervention group | 0,03 | 0,12 | 0,779 |
| | False news | Intercept | −0,09 | 0,06 | 0,106 |
| | | Posttest | 0,11 | 0,06 | 0,050 |
| | | Intervention group | 0,07 | 0,11 | 0,564 |
| | | Posttest x Intervention group | 0,17 | 0,11 | 0,144 |
| Psychology | True news | Intercept | −0,17 | 0,07 | 0,014 |
| | | Posttest | 0,27 | 0,05 | 0,000 |
| | | Intervention group | 0,14 | 0,22 | 0,518 |
| | | Posttest x Intervention group | −0,09 | 0,15 | 0,540 |
| | False news | Intercept | −0,07 | 0,05 | 0,168 |
| | | Posttest | 0,15 | 0,05 | 0,004 |
| | | Intervention group | −0,02 | 0,16 | 0,909 |
| | | Posttest x Intervention group | −0,01 | 0,15 | 0,959 |
| Natural sciences | True news | Intercept | −0,15 | 0,07 | 0,026 |
| | | Posttest | 0,25 | 0,05 | 0,000 |
| | | Intervention group | −0,14 | 0,28 | 0,628 |
| | | Posttest x Intervention group | 0,18 | 0,26 | 0,500 |
| | False news | Intercept | −0,07 | 0,05 | 0,148 |
| | | Posttest | 0,15 | 0,05 | 0,004 |
| | | Intervention group | −0,05 | 0,25 | 0,826 |
| | | Posttest x Intervention group | 0,16 | 0,25 | 0,515 |
| Arts | True news | Intercept | −0,16 | 0,07 | 0,021 |
| | | Posttest | 0,27 | 0,05 | 0,000 |
| | | Intervention group | 0,00 | 0,30 | 0,994 |
| | | Posttest x Intervention group | −0,23 | 0,24 | 0,333 |
| | False news | Intercept | −0,08 | 0,05 | 0,135 |
| | | Posttest | 0,15 | 0,05 | 0,002 |
| | | Intervention group | 0,02 | 0,24 | 0,917 |
| | | Posttest x Intervention group | −0,02 | 0,23 | 0,916 |

## Appendix F

**Table F1.** Table 5 (regression results for News Evaluator) with p-values.

| | Post 1 (true) | p-value | Post 2 (false) | p-value | Use of digital tools | p-value |
|---|---|---|---|---|---|---|
| Intercept | −0.05 (0.12) | 0.662 | −0.16 (0.11) | 0.165 | −1.98*** (0.50) | 0.000 |
| Posttest | 0.04 (0.13) | 0.750 | 0.13 (0.14) | 0.355 | 0.08 (0.49) | 0.875 |
| News Evaluator | −0.08 (0.15) | 0.558 | 0.16 (0.14) | 0.229 | −0.45 (0.56) | 0.423 |
| Posttest x News Evaluator | 0.31 (0.16) | 0.056 | −0.07 (0.17) | 0.689 | 0.11 (0.60) | 0.859 |

## Supporting information

**S1 File. This file includes links to the preregistration, teaching materials used in the intervention, data, and additional methodological details.**
(DOCX)

## Author contributions

**Conceptualization:** Thomas Nygren.

**Data curation:** Thomas Nygren, Evgenia Efimova.

**Formal analysis:** Thomas Nygren, Evgenia Efimova.

**Funding acquisition:** Thomas Nygren.

**Investigation:** Thomas Nygren.

**Methodology:** Thomas Nygren.

**Project administration:** Thomas Nygren.

**Resources:** Thomas Nygren.

**Supervision:** Thomas Nygren.

**Visualization:** Evgenia Efimova.

**Writing – original draft:** Thomas Nygren, Evgenia Efimova.

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
