## [Decision Letter · Decision Letter 0]

Dear Dr. Nygren,

Thank you for submitting your manuscript to PLOS ONE. After careful consideration, we feel that it has merit but does not fully meet PLOS ONE’s publication criteria as it currently stands. Therefore, we invite you to submit a revised version of the manuscript that addresses the points raised during the review process.

We look forward to receiving your revised manuscript.

Kind regards,

Prof. Anat Gesser-Edelsburg, Ph.D.

Academic Editor

PLOS ONE

Journal Requirements:

“This study was funded by the Swedish Institute for Educational Research, grant no 2020-00009. https://www.skolfi.se/other-languages/english/ Funding awarded TN”

4. Please note that your Data Availability Statement is currently a direct link can’t access each database. If your manuscript is accepted for publication, you will be asked to provide these details on a very short timeline. We therefore suggest that you provide this information now, though we will not hold up the peer review process if you are unable.

Reviewers' comments:

Reviewer's Responses to Questions

**Comments to the Author**

1. Is the manuscript technically sound, and do the data support the conclusions?

Reviewer #1: Yes

Reviewer #2: Yes

Reviewer #3: Yes

2. Has the statistical analysis been performed appropriately and rigorously?

Reviewer #1: Yes

Reviewer #2: Yes

Reviewer #3: Yes

3. Have the authors made all data underlying the findings in their manuscript fully available?

Reviewer #1: Yes

Reviewer #2: Yes

Reviewer #3: Yes

4. Is the manuscript presented in an intelligible fashion and written in standard English?

Reviewer #1: Yes

Reviewer #2: Yes

Reviewer #3: Yes

Reviewer #1: This paper investigates the effects of interventions aimed at improving students' ability to identify and evaluate misinformation in upper secondary education in Sweden. The study uses a longitudinal design (with pre-test and post-test) to assess both the immediate and delayed impact of instructional interventions focused on critical thinking and source evaluation. The use of digital tools is taken into account.

The paper has many merits:

- it addresses a timely and socially critical topic—combating misinformation—through a rigorous empirical approach. Education is considered one of the main 'antidote' agains misinformation, and we need many rigorous studies on the long term effect of 'inoculating' small doses of misinformation to students, with how to correct misleading information;

- to my understanding, the analysis is solid, methodologically sound, and supported by appropriate statistical tools.

- the experimental design is clearly described, including procedures and instruments, allowing for full or partial replicability in other schools or countries.

- limitations are transparently acknowledged, strengthening the credibility of the conclusions.

However, some of the data tables are dense and require significant effort to interpret; the inclusion of additional summarizing charts or simplified visualizations would greatly enhance clarity and reader engagement. The paper already contains some basic graphs, but some more effective visualization or info graphic that can effectively represent the most important and technical aspects can be designed.

Reviewer #2: This is a very solid report of a good piece of research. It is very well anchored clearly written, present a good argument and a very clear account of the study, the analysis and the outcomes. It’s very interesting to read and has a good discussion.

The reviewer copy notes that an update can be downloaded, so I have reviewed the updated version. However, there is nothing that explains what has changed, which isn’t obvious from a quick look through.

The main points I’d like to note concern fairly specific aspects of clarity in one or two places, and the overall argument.

On p.14 it’s mentioned that the control group were more advanced and with a stronger academic background. This isn’t further commented on, but might it not be noteworthy in some respects? For example, is it surprising that there are no significant differences (p.18) between the control group and the experimental groups at pre-test?

A recurrent point, which is very interesting, is that certain factors seem to enhance ability (of certain groups) to recognise credible news, whereas others enhance ability to detect misinformation. It might be useful to discuss this distinction a little further. One might be tempted to think that, if a news item is either accurate or not, failing to recognise it as accurate amounts to recognising it as not accurate. Perhaps there is a state where judgement on an item is suspended, or just never made, or something like that, but this isn’t discussed. Related to this, there’s a tendency to conflate “accurate”, “credible” and even “true”, which may not be quite right (although perhaps it’s a workable approximation).

There’s an observation in several places, e.g. p.30 that “higher AOT scores correlated with scepticism toward both true and false news, suggesting that instead of improving discernment, AOT may be linked to a generalised distrust of information“. Perhaps the subtlety here is too much for even senior secondary students, but one might have hoped there was room for a notion of confidence in sources. No source or news item, surely, should ever be accepted as infallible, so that its truth is simply a given! But it may be more credible, so that one can have greater confidence in the likelihood of its being true. Is it just that the instrument used in the study was not capable of detecting levels of confidence (or degrees of scepticism), for instance? (From this perspective, a “general scepticism” is an entirely appropriate and necessary part of analytical thinking, and it’s not at all the same thing as “indiscriminate doubt”.)

In many ways the major message of the paper is that the effects of the kind of intervention studied tend not to be long-term or persistent. There seem to be two things to say about this.

(1) It’s not as clear as it might be, at the outset, that this is a focus of the study, at least in the sense that it’s not represented explicitly in any of the hypotheses. Should they all be read as including “as measured after the lapse of <some time="">”? (Related to this, only the first three hypotheses are explicitly stated anyway, but perhaps the others are clear enough.)

(2) There is little if any discussion in the background section about this particular point. Hence, where would the hypothesis come from? Much, if not most of the existing literature, across many areas of education and learning (at least as I would read it) suggests that one-off or short-term interventions rarely (or only in particular facilitating circumstances) lead to sustained behavioural or cognitive changes. Hence one would be surprised if these interventions were any different. Surely it’s to be expected that meaningful change requires attitudes and practices to be inculcated over a significant period of time?

Very minor typos:

p.5 para.1 "in as" should be just one of these, probably "in"?

p.9 missing full stop at the end of H2.

p.9 last para. -- capitalisation of "Bad News" (twice).

p.13 again, capitalisation of "Bad News".</some>

Reviewer #3: The authors are thanked for their labor-intensive and engaging research. The study is highly significant in that it provides insights into the long-term effects of various interventions. Although the inclusion of different groups and interventions occasionally makes it challenging to follow the text, the research design stands out as one of the study’s major strengths.

Some specific areas of the text are suggested for revision, as outlined below:

- While the inclusion of hypotheses is appreciated, it is also recommended that the research questions guiding the study be explicitly articulated in the text. Positioning the research questions just before the hypotheses could enhance the clarity of the study's framework and objectives.

- In Table 2, for Analysis 4 (Factors of identifying true and false news), the dependent variable is described as “Mean reliability ratings for true news and false news, from one to seven.” It might be helpful to clarify whether this rating is consistent across all independent variables, or if it varies (e.g., 1 to 5 or 1 to 10), as this could influence interpretation.

- Is it possible that there might be an error in the table numbers mentioned in the second paragraph under the “Analysis” section? In the sentence: “…As shown by Tables 4 and 5, there are no significant differences between the control and experimental groups in the pre-test.”

- In the first paragraph under the heading “Fact-checking skills intervention (News Evaluator)” (page 20), the following is stated: “…However, there might be some indication of a positive effect for the first Facebook post (p = 0.056)…”. Is it possible to locate this reported p-value in any of the tables in the main text or appendix?

**Do you want your identity to be public for this peer review?** For information about this choice, including consent withdrawal, please see our Privacy Policy

Reviewer #1: No

Reviewer #2: No

Reviewer #3: No

---

## [Author Response · Author response to Decision Letter 1]

2 Jun 2025

Response to reviewers

Reviewer #1: This paper investigates the effects of interventions aimed at improving students' ability to identify and evaluate misinformation in upper secondary education in Sweden. The study uses a longitudinal design (with pre-test and post-test) to assess both the immediate and delayed impact of instructional interventions focused on critical thinking and source evaluation. The use of digital tools is taken into account.

The paper has many merits:

- it addresses a timely and socially critical topic—combating misinformation—through a rigorous empirical approach. Education is considered one of the main 'antidote' agains misinformation, and we need many rigorous studies on the long term effect of 'inoculating' small doses of misinformation to students, with how to correct misleading information;

- to my understanding, the analysis is solid, methodologically sound, and supported by appropriate statistical tools.

- the experimental design is clearly described, including procedures and instruments, allowing for full or partial replicability in other schools or countries.

- limitations are transparently acknowledged, strengthening the credibility of the conclusions.

However, some of the data tables are dense and require significant effort to interpret; the inclusion of additional summarizing charts or simplified visualizations would greatly enhance clarity and reader engagement. The paper already contains some basic graphs, but some more effective visualization or info graphic that can effectively represent the most important and technical aspects can be designed.

Response: We appreciate this thoughtful suggestion and fully agree that enhanced visualisations can support clarity and engagement. We have considered adding simplified figures (e.g., summary bar charts of pre- and post-test results for each group), and we experimented with several options. However, due to the complexity of our design—multiple overlapping interventions, several outcome variables, and mixed model estimates across nested data—it proved difficult to produce clear and accurate summary visualisations without risking oversimplification or misrepresentation of the findings.

Reviewer #2: This is a very solid report of a good piece of research. It is very well anchored clearly written, present a good argument and a very clear account of the study, the analysis and the outcomes. It’s very interesting to read and has a good discussion.

The reviewer copy notes that an update can be downloaded, so I have reviewed the updated version. However, there is nothing that explains what has changed, which isn’t obvious from a quick look through.

The main points I’d like to note concern fairly specific aspects of clarity in one or two places, and the overall argument.

On p.14 it’s mentioned that the control group were more advanced and with a stronger academic background. This isn’t further commented on, but might it not be noteworthy in some respects? For example, is it surprising that there are no significant differences (p.18) between the control group and the experimental groups at pre-test?

A recurrent point, which is very interesting, is that certain factors seem to enhance ability (of certain groups) to recognise credible news, whereas others enhance ability to detect misinformation. It might be useful to discuss this distinction a little further. One might be tempted to think that, if a news item is either accurate or not, failing to recognise it as accurate amounts to recognising it as not accurate. Perhaps there is a state where judgement on an item is suspended, or just never made, or something like that, but this isn’t discussed. Related to this, there’s a tendency to conflate “accurate”, “credible” and even “true”, which may not be quite right (although perhaps it’s a workable approximation).

There’s an observation in several places, e.g. p.30 that “higher AOT scores correlated with scepticism toward both true and false news, suggesting that instead of improving discernment, AOT may be linked to a generalised distrust of information“. Perhaps the subtlety here is too much for even senior secondary students, but one might have hoped there was room for a notion of confidence in sources. No source or news item, surely, should ever be accepted as infallible, so that its truth is simply a given! But it may be more credible, so that one can have greater confidence in the likelihood of its being true. Is it just that the instrument used in the study was not capable of detecting levels of confidence (or degrees of scepticism), for instance? (From this perspective, a “general scepticism” is an entirely appropriate and necessary part of analytical thinking, and it’s not at all the same thing as “indiscriminate doubt”.)

In many ways the major message of the paper is that the effects of the kind of intervention studied tend not to be long-term or persistent. There seem to be two things to say about this.

(1) It’s not as clear as it might be, at the outset, that this is a focus of the study, at least in the sense that it’s not represented explicitly in any of the hypotheses. Should they all be read as including “as measured after the lapse of ”? (Related to this, only the first three hypotheses are explicitly stated anyway, but perhaps the others are clear enough.)

(2) There is little if any discussion in the background section about this particular point. Hence, where would the hypothesis come from? Much, if not most of the existing literature, across many areas of education and learning (at least as I would read it) suggests that one-off or short-term interventions rarely (or only in particular facilitating circumstances) lead to sustained behavioural or cognitive changes. Hence one would be surprised if these interventions were any different. Surely it’s to be expected that meaningful change requires attitudes and practices to be inculcated over a significant period of time?

Very minor typos:

p.5 para.1 "in as" should be just one of these, probably "in"?

p.9 missing full stop at the end of H2.

p.9 last para. -- capitalisation of "Bad News" (twice).

p.13 again, capitalisation of "Bad News".

Response: We now highlight that control group students came from more advanced, theoretical tracks but still showed no pre-test advantage. In the discussion, we reflect on how academic placement does not guarantee digital source evaluation skills.

We added clarifications emphasizing the difference between perceived reliability and factual accuracy.

We now include a more nuanced interpretation of AOT results.

We revised the introduction and prefaced the hypotheses with a sentence clarifying the focus on long-term outcomes.

Even if the challenge of long-term effects is highlighted in educational literature (as already mentioned and now better underlined in the paper), most studies in the misinformation field do short interventions, and this is why it is necessary to do this kind of study to provide evidence of potential and challenges.

Reviewer #3: The authors are thanked for their labor-intensive and engaging research. The study is highly significant in that it provides insights into the long-term effects of various interventions. Although the inclusion of different groups and interventions occasionally makes it challenging to follow the text, the research design stands out as one of the study’s major strengths.

Some specific areas of the text are suggested for revision, as outlined below:

- While the inclusion of hypotheses is appreciated, it is also recommended that the research questions guiding the study be explicitly articulated in the text. Positioning the research questions just before the hypotheses could enhance the clarity of the study's framework and objectives.

- In Table 2, for Analysis 4 (Factors of identifying true and false news), the dependent variable is described as “Mean reliability ratings for true news and false news, from one to seven.” It might be helpful to clarify whether this rating is consistent across all independent variables, or if it varies (e.g., 1 to 5 or 1 to 10), as this could influence interpretation.

- Is it possible that there might be an error in the table numbers mentioned in the second paragraph under the “Analysis” section? In the sentence: “…As shown by Tables 4 and 5, there are no significant differences between the control and experimental groups in the pre-test.”

- In the first paragraph under the heading “Fact-checking skills intervention (News Evaluator)” (page 20), the following is stated: “…However, there might be some indication of a positive effect for the first Facebook post (p = 0.056)…”. Is it possible to locate this reported p-value in any of the tables in the main text or appendix?

Response: We have added two guiding research questions before the hypotheses.

We added scales of the independent variables in Table 2. Also, we present regression results in the analysis section in standardized form.

Thank you very much, it was an error indeed.

An extended table with p-values was added as Appendix F.

---

## [Editor Report · Decision Letter 1]

Investigating the Long-Term Impact of Misinformation Interventions in Upper Secondary Education

PONE-D-25-15019R1

Dear Dr. Nygren,

We’re pleased to inform you that your manuscript has been judged scientifically suitable for publication and will be formally accepted for publication once it meets all outstanding technical requirements.

Kind regards,

Prof. Anat Gesser-Edelsburg, Ph.D.

Academic Editor

PLOS ONE
---

## [Editor Report · Acceptance letter]

PONE-D-25-15019R1

PLOS ONE

Dear Dr. Nygren,

I'm pleased to inform you that your manuscript has been deemed suitable for publication in PLOS ONE. Congratulations! Your manuscript is now being handed over to our production team.

Kind regards,

on behalf of

Prof. Anat Gesser-Edelsburg

Academic Editor

PLOS ONE